# Prognostic Factors for Post-Recurrence Survival in Stage II and III Colorectal Carcinoma Patients

**DOI:** 10.3390/medicina57101108

**Published:** 2021-10-15

**Authors:** Neda Nikolic, Davorin Radosavljevic, Dusica Gavrilovic, Vladimir Nikolic, Nemanja Stanic, Jelena Spasic, Tamara Cacev, Sergi Castellvi-Bel, Milena Cavic, Goran Jankovic

**Affiliations:** 1Institute for Oncology and Radiology of Serbia, 11000 Belgrade, Serbia; davorr@ncrc.ac.rs (D.R.); duca@ncrc.ac.rs (D.G.); vladimir.nikolic@ncrc.ac.rs (V.N.); nemanja.s.stanic@gmail.com (N.S.); jelena.spasic@ncrc.ac.rs (J.S.); milena.cavic@ncrc.ac.rs (M.C.); 2Division of Molecular Medicine, Rudjer Boskovic Institute, 10000 Zagreb, Croatia; tcacev@irb.hr; 3Centro de Investigación Biomédica en Red de Enfermedades Hepáticas y Digestivas (CIBERehd), Gastroenterology Department, Institut d’Investigacions Biomèdiques August Pi i Sunyer (IDIBAPS), Hospital Clínic, University of Barcelona, 08007 Barcelona, Spain; SBEL@clinic.cat; 4Clinic for Gastroenterology and Hepatology, Clinical Centre of Serbia, 11000 Belgrade, Serbia; goran.jankovic@kcs.ac.rs

**Keywords:** post-recurrence survival, colorectal cancer, prognostic factor

## Abstract

*Background and objectives*: This study aimed to evaluate prognostic factors for post-recurrence survival in local and locally advanced colorectal cancer patients. *Materials and Methods:* A total of 273 patients with stage III and high-risk stage II colorectal cancer were prospectively enrolled. All patients underwent operative treatment of the primary tumor and adjuvant fluorouracil-based chemotherapy. *Results:* Over the three-year period (2008–2010), a cohort of 273 patients with stage III and high-risk stage II colorectal cancer had been screened. During follow up, 105 (38.5%) patients had disease recurrence. Survival rates 1-, 3- and 5-year after recurrence were 53.9, 18.2 and 6.5%, respectively, and the median post-recurrence survival time was 13 months. Survival analysis showed that age at diagnosis (*p* < 0.01), gender (*p* < 0.05), elevated postoperative Ca19-9 (*p* < 0.01), tumor histology (adenocarcinoma vs. mucinous vs. signet ring tumors, *p* < 0.01) and tumor stage (II vs. III, *p* < 0.05) had a significant influence on post-recurrence survival. Recurrence interval and metastatic site were not related to survival following recurrence. Multivariate analysis showed that older age (HR 2.43), mucinous tumors (HR 1.51) and tumors expressing Ca19-9 at baseline (HR 3.51) were independently associated with survival following recurrence. *Conclusions:* Baseline patient and tumor characteristics largely predicted patient outcomes after disease recurrence. Recurrence intervals in local and locally advanced colorectal cancer were not found to be prognostic factors for post-recurrence survival. Older age, male gender, stage III and mucinous histology were poor prognostic factors after the disease had recurred. Stage II patients had remarkable post-recurrence survival compared to stage III patients.

## 1. Introduction

Colorectal cancer (CRC) is the third most commonly diagnosed cancer and one of the leading causes of cancer-related deaths worldwide [1]. In recent years, there has been a steady decline in its overall incidence, due to the implementation of population-based screening programs and rising understanding of the necessity for a healthy lifestyle. Patient survival has also increased due to developments in early diagnosis, personalized therapy and broader knowledge of tumor biology [2]. However, in developing countries, it remains a significant public health burden. In 2018, 4646 new colorectal cancer cases had been diagnosed in Serbia, and 2591 deaths caused by the same cancer had been recorded [3], marking it as the second most common cause of cancer-related morbidity and mortality. 

Approximately 22% of CRCs are metastatic at initial diagnosis, with a relative 5-year survival rate of 14%, compared to a 71 and 90% survival rate in those with regional and localized CRC, respectively [2]. A significant number of patients can be cured with surgical resection only [4]. Curative surgery consists of partial colectomy and resection of at least 12 lymph nodes (LNs). Once surgery has been performed, a precise pathohistological report is crucial to stratify the risk of relapse and consequently tailor the individual adjuvant treatment for each patient. The report should consist of bowel wall infiltration (pT-status), number of affected LNs (pN-status), resection margin clearance, differentiation degree, lymphovascular/perineural invasion and microsatellite instability (MSI) status [5,6]. Novel tools for molecular characterization such as next generation sequencing (NGS) and markers such as BRAF (B-Raf proto-oncogene, serine/threonine kinase), RAS (Rat sarcoma virus), CDX2 (Caudal Type Homeobox 2) genes have shown a high prognostic value but have still not been validated for treatment guidance [7,8,9,10]. Circulating tumor DNA (ctDNA) is a new rapidly emerging field that may be a good surrogate of minimal residual disease in CRC, which may guide treatment in stage II and III disease in the future [11]. The gene expression–based consensus molecular subtypes (CMS) has been introduced in 2015, as an effort of the international consortium to resolve inconsistencies among the reported gene expression–based CRC classifications. Four robust transcriptome-based subtypes were established: CMS1 (microsatellite instability immune, 14%); CMS2 (canonical, 37%); CMS3 (metabolic, 13%); CMS4 (mesenchymal, 23%) and samples with mixed features (13%) that possibly represent a transition phenotype or intratumoral heterogeneity. Each subtype displays different pathological and genetic signatures. Based on these features, therapeutic stratification for individual patients may be designed, which may ultimately lead to improved therapeutic outcomes [12]. The search for the ideal biomarker for outcome prediction is ongoing and numerous studies so far addressed this issue [13,14].

When diagnosed at a stage when curative resection is possible, the addition of adjuvant chemotherapy provides significant disease-free survival gain that translates into a long-term overall survival benefit [15]. Stage III and high-risk stage II are diagnosed in more than half of patients and curative surgical resection followed by adjuvant chemotherapy is a standard clinical approach. Tumor recurrence is a major obstacle in the cure and long-term survival of patients. Recurrence rates have been reported to be up to 40% depending on the stage [16]. The probability of recurrence of local and locally advanced CRC is commonly based on predictors present at diagnosis. Prognosis and survival after recurrence of the disease are typically based on tumor and patient characteristics at the time of diagnosis of advanced disease. These include the metastatic site, probability of surgical resection, performance status, and disease burden. Management of advanced CRC is mainly oriented on the recognition of advanced CRC patients as a heterogeneous patient population and, consequently, to a “clinical and molecular” personalized therapy [17,18]. Despite aggressive and multimodal treatments, most recurrent patients have a low probability of being cured.

Relatively few studies on prognostic factors for post-recurrence survival (PRS) have been reported. In this study, we aimed to determine whether initial patient characteristics or features of the primary tumor could be relevant factors in predicting future clinical behavior after disease recurrence in patients who had previously undergone complete resection of stage II or III tumors and adjuvant chemotherapy.

## 2. Materials and Methods

### 2.1. Patient Characteristics

Between January 2008 and December 2010, 1187 patients with CRC were treated at the Institute for Oncology and Radiology of Serbia. A total of 273 patients with high-risk stage II and stage III CRC who underwent a complete resection of the primary tumor and were treated with adjuvant chemotherapy had been identified. Among them, 249 (91.2%) patients had colon and 24 (8.8%) proximal rectal cancer. Rectal cancer patients were not subjected to a preoperative treatment due to the localization of the primary tumor. Collected data contain patient characteristics (ECOG PS-Eastern Cooperative Oncology group performance status), histopathological reports, recurrence intervals, site of first recurrence, and date of last visit and death. All data were collected from patient medical records or over the telephone, and their association with post-recurrence survival was analyzed. This study was approved by the Institutional Ethics Committee and by the Ethics Committee and Review Board of the University of Belgrade, Serbia (Protocol number 1322/X-40). Signed informed consent was obtained from each patient prior to participation in the study. 

Patients were staged according to the 7th TNM (tumor-node-metastasis) staging, proposed by the UICC (Union International for cancer control UICC) [19]. Histologic subtypes of CRC were determined according to World Health Organization classification [19,20]. High-risk stage II disease was defined as node-negative disease with high-risk features: T4, lymphovascular invasion, grade 3, intestinal perforation and less than 12 nodes examined in the specimen [20]. For all patients, serum concentrations of CEA (carcinoma embryonic antigen) and Ca19-9 (carbohydrate antigen 19-9) were assessed at baseline (before starting adjuvant chemotherapy).

### 2.2. Treatment and Follow-Up

All patients were treated with adjuvant chemotherapy. The adjuvant chemotherapy consisted of fluorouracil-based chemotherapy, Mayo regimen: 5-fluorouracil 425 mg/m^2^ + Leucovorin 25 mg/m^2^, D1-5, Q4W, 6 cycles or Capecitabine, 2500 mg/m^2^, D1-14, Q3W, 8 cycles, according to the reimbursement list of the Serbian Health Insurance Fund. During the follow-up period, blood examinations (including serum tumor markers CEA, Ca19-9), chest X-ray or chest/abdomen/pelvis CT scan or abdominal ultrasonography were performed every three months for the first two years after resection and every six months thereafter, according to European Society for Medical Oncology (ESMO) and national guidelines [21]. Patients were followed up until disease recurrence or death from any cause. The criteria for establishing a recurrent disease included radiographic evidence of disease recurrence, clinically palpable mass or histopathology confirmation, if applicable. Recurrences were divided into three groups: local, distant and combined. Local recurrence was defined as recurrence within or contiguous to the previously treated tumor bed and distant recurrence was defined as disease spread outside the primary tumor basin. Recurrence in local lymph nodes was considered as distal spread.

### 2.3. Salvage Therapies

According to the type of salvage treatment after disease recurrence, patients were divided into the following groups: no systemic treatment (patients who were treated with best supportive care (BCS); local radiotherapy (RT) to the bone or brain was allowed), surgery only (patients who were amenable for radical surgery), palliative chemotherapy with and without biologics (patient who were not candidates for curative surgery or radiotherapy) or combination therapy (patients who were treated with combination therapy that included systemic chemotherapy with and without biologics, surgery, and radiotherapy).

### 2.4. Statistical Analysis

Statistical analysis was performed using the program R (version 3.3.2 (31 October 2016)—“Sincere Pumpkin Patch”; Copyright (C) 2016 The R Foundation for Statistical Computing; Platform: x86_64-w64-mingw32/× 64 (64-bit); downloaded: 21 January 2017) [22]. Curves of probabilities for post-recurrence survival were constructed using the Kaplan–Meier product-limit method; the median of survival analysis with corresponding 95% CI was used for description and the log-rank test was utilized for testing differences between curves. All reported *p* values were two-sided, with *p* < 0.05 denoting statistical significance. Values over 0.05 were denoted as not statisticaly significant (ns-not significant). Univariate and multivariate Cox proportional hazard regression models were used; the hazard ratio (HR) with the corresponding 95% CI (confidence interval) were utilized for description; the Wald and Likelihood ratio test was utilized for statistical testing. 

## 3. Results

### 3.1. Patient Characteristics

Out of the 273 patients who underwent curative resection followed by adjuvant chemotherapy for high-risk stage II and III CRC, 105 patients experienced disease recurrence (38.5%). Median follow-up time was 66 months (range 3–148 months), and the median time to initial recurrence was 17 months. Most patients experienced recurrence in the first year of follow-up (39 patients, 37%) and 68.6% of recurrences were diagnosed in the first 2 years after surgery. Thirteen patients had a recurrence more than four years after the surgery (12.4%). Post-recurrence survival for the whole group is presented in Figure 1. 

### 3.2. Post-Recurrence Survival

The 1-, 3- and 5-year survival rates of all patients after recurrence were 51.04, 18.24, 6.51%, respectively. The median post-recurrence survival time was 13 months (11–17, 95% CI). The cumulative percentage of yearly post-recurrence survival during follow-up with 95% CI is presented in Table 1.

Differences in post-recurrence survival according to patient characteristics, histopathological findings, pathologic stage, recurrence pattern and time to recurrence is presented in Table 2.

Survival analysis showed that age at diagnosis, gender, elevated postoperative Ca19-9, tumor histology and tumor stage had a statistically significant influence on post-recurrence survival. Survival following recurrence was strongly affected by age at diagnosis, showing worse outcomes in patients of 70 years or older, who had only 7 months of post-recurrence survival compared with 18 months in patients under 70 (*p* < 0.01; Table 2). An influence of gender was detected, with a worse outcome for male patients (median 17 vs. 9.5 months, respectively, *p* < 0.05, Table 2). Histopathology was strongly related to survival following recurrence, indicating worse outcomes for mucinous and signet ring tumors. Median post-recurrence survival was 15 vs. 8.5 vs. 4 months for adenocarcinoma, mucinous and signet ring tumors, respectively (*p* < 0.01, Table 2). Patients who initially had a high level of postoperative Ca19-9 (≥37) had worse survival following recurrence than patients with normal Ca19-9 levels at baseline (*p* < 0.01, Table 2).

In patients with stage II disease, the 5-year post-recurrence survival rate reached 27.8% and median survival was 47 months, whilst for stage III tumors the 5-year post-recurrence survival rate was 4.7% and median survival was 13 months (*p* < 0.05). Time from diagnosis to recurrence and site of metastasis were not related to survival following recurrence. Kaplan–Meier plots for post-recurrence survival according to investigated variables are shown in Figure 2.

Parameters that were statistically significant in the log-rank test were further assessed using the univariate and multivariate Cox regression analysis. Multivariate analysis showed that (95% CI): older age (HR 2.43, range: 1.55–3.81), signet ring carcinoma (HR 9.69, range: 2.23–41.9), mucinous tumors (HR 1.51, range 0.73–3.10) and tumors expressing Ca19-9 at baseline (HR 3.51, range 1.68–7.37) were independently associated with survival following recurrence (Table 3).

### 3.3. Salvage Therapies

Out of 105 patients with recurrent disease, 22.8% (24) were not amenable to systemic treatment at diagnosis of metastatic disease. All of them were treated with best supportive care; in four patients, radiotherapy to the bone was the only treatment, and one patient had radiotherapy to the brain. The type of salvage treatment and their influence on survival are presented in Table 4 and Figure 3. The longest survival was observed in the group who were treated with upfront resection, as the only treatment modality (38.5 months). Patients who were treated with combined treatment and chemotherapy only had a post-recurrence survival of 24 and 13 months, respectively. Patients who were not amenable to systemic treatment had the worst median post-recurrence survival of only 3 months.

## 4. Discussion

In this study, the risk factors affecting post-recurrence survival of patients with local and locally advanced CRC who underwent surgical resection followed by adjuvant chemotherapy were analyzed. CRC is a growing public health problem in Serbia and worldwide, yet there is scarce available data on prognostic factors and patient outcomes. The obtained data indicated that patients’ age (>70 years) as well as mucinous histology were independent predictors of post-recurrence survival. The influence of the time to recurrence on post-recurrence survival was not confirmed. Stage at diagnosis was a prognostic factor for post-recurrence survival, but not an independent one. 

Older patients comprise a majority of diagnosed CRC, as 60% of them are older than 70 years at the time of diagnosis. Patients older than 75 years also account for half of CRC deaths [2]. In general, survival of cancer patients usually decreases with age [23,24]. Survival of older patients is not only limited by their primary disease itself, but rather by their comorbidities, frailty, and psychological and social care issues. In their study, Doat et al. concluded that with increasing age, overall survival, and thus follow-up, decreases—as does the intensity of cancer treatment [25]. Our group comprised of 63.8% patients older than 70 years at the time of diagnosis of primary CRC. All of them were treated with adjuvant chemotherapy. When disease recurred, older age was found to be an independent poor prognostic factor for survival following recurrence in the multivariate analysis. Previously published large cohort studies had found an inverse relationship between age and survival [26,27]. This fact does not come as a surprise; younger patients are expected to have longer survival times, as they have fewer comorbidities, lower probabilities of dying from other causes and are likely to receive more aggressive treatments [28].

Differences in CRC survival relating to patients’ gender have been observed in previously published studies. A consistent higher rate of survival in female patients has been reported [29,30,31,32]. The difference is most prominent in young and middle-aged patients [29]. The observed trend might be attributed to sex hormones, higher levels of health awareness and potential differences in risk behaviors (smoking, physical inactivity, etc.) [32]. Our data confirmed gender-dependent differences in post-recurrence survival. It was not confirmed as an independent predictor in the multivariate model—the rationale for which might be the lack of age-stratification. Further research on these aspects should be pursued.

Different origins of the right and left side of the colon lead to differences in clinical behavior, and chemo and biological therapy response according to the side of the primary tumor [33]. The influence of the sidedness of the primary tumor in prognosis is still debatable and is largely impacted by the stage of the disease [34]. Our data showed a trend of longer post-recurrence survival in patients with left-sided tumors, who had a survival time 6 months longer compared to patients with right-sided tumors. The observed result failed to reach statistical significance, probably due to the small sample size. In stage IV CRC, the side of the primary tumor is an important determinant and should be taken into account when analyzing the prognosis of individual patients.

Previous studies identified Ca19-9 as a prognostic factor in CRC [35,36]. Elevated baseline Ca19-9 was attributed to a high risk of postoperative metastasis, which contributes to a worse survival rate. This study confirmed the prognostic significance of Ca19-9, even for post-recurrence survival. Although evaluation of postoperative Ca19-9 levels is not recommended by the prominent guidelines, the clinical value of Ca19-9 measurement is undoubted. It showed a strong prognostic influence in our study and might be proposed as a minimally-invasive, low-cost prognostic marker in this setting.

Mucinous adenocarcinomas are a rare and distinct entity of CRC. Conflicting data on prognosis and survival of signet ring cell and mucinous adenocarcinomas exists, since most of the data comes from retrospective series [37]. In this study group, the frequencies of mucinous and signet ring histology tumors were as expected, at 9.5% and less than 2%, respectively. Non-adenocarcinoma histology was confirmed as an independent factor for post-recurrence prognosis (HR for mucinous 1.506; HR for signet ring 9.687). This observation can be attributed to clinical and genetic characteristics of mucinous cancers and poor responses to palliative systemic therapy compared to standard adenocarcinoma [38]. 

Stage remains one of the most important prognostic factors for survival of CRC patients. Regarding the influence of stage at diagnosis, patients with stage II had an outstanding survival following recurrence that reached 47 months. In stage III, median post-recurrence survival was only 13 months (*p* < 0.05). Although this difference was clinically and statistically significant, it did not prove to be an independent factor for post-recurrence survival in the multivariate analysis. The reason might lie in the small sample size of stage II patients, as less than 10% of analyzed patients had stage II and tumor recurrence.

O’Connell et al. explored survival following recurrences in patients from ACCENT database in 2008 [39]. They concluded that stage of the disease and recurrence interval were independent predictors for post-recurrence survival. Particularly, in patients with initial stage III disease, time from randomization was significantly associated with survival following recurrence, but not in patients with initial stage II disease. Other retrospective observational studies have confirmed that patients who had shorter DFS (<12 months) have poorer outcomes [40,41]. Recurrence in the first two years of follow up is a poor prognostic factor as well [42]. These data are in accordance with the concept that the biology of fast recurring cancers contributes to their aggressiveness and renders them less likely to be impacted by adjuvant chemotherapy. However, this was not reflected in the results of this study. Almost 40% of recurrences were diagnosed in the first year of follow up, but data do not support the poor outcome of this group. Early recurrence had no impact on post-recurrence survival. This result could only be explained with regard to missing stage stratification. Stage II tumors tend to have longer post-recurrence survival in general. Stratification based on stages at diagnosis could provide better insight into this matter. Broadbridge et al. in 2013 have explored the prognosis of late recurring cancers (more than 5 years from diagnosis) and found no survival differences between early and late recurrences of colon cancer [43]. Expected ‘indolent’ biology, and thus better survival in late recurring cancers, could not be confirmed. Late recurrences were more common in older patients, left-sided colon cancer and stage II. We reached similar results in our research. Late recurrence did not lead to prolonged post-recurrence survival. These results clearly point out that the recurrence interval could not be used as a single independent prognostic determinant of individual patients.

When a diagnosis of metastatic disease is reached, further treatment options are explored depending on patient- and disease-related relevant factors. Patient-related factors taken into consideration are performance status, symptom burden, comorbidities, patient expectation and motivation. Disease-related factors combine patterns of tumor biology and clinical presentation of the disease. The treatment strategy is established by defining a treatment aim and the possibility of a multimodal approach. In our group, 22.8% of patients didn’t receive any systemic treatment and this group had the worst survival, as expected. The best post-recurrence survival was shown for patients who were treated with upfront resection where that was the only treatment modality throughout the course of their metastatic disease. This clinical setting was possible due to ‘good’ tumor biology and metastases that were limited in number and size. Good long-term survival was also achieved in the group of patients who were treated with combined therapies—surgery, chemotherapy (with or without biologics) and radiotherapy. In the group of patients who received systemic treatment, the shortest survival was seen in patients with non-resectable metastatic disease and with palliative treatment intentions.

To the best of our knowledge, this is the first study of this type performed on the Slavic population in the Western Balkan area which is combined with results on the genetic background of patients [44,45,46] might be useful for future meta-analyses, as various population-based factors might also be significant.

## 5. Conclusions

Baseline patient and tumor characteristics largely predict a patient’s outcome after disease recurrence. Older age, male gender and mucinous histology were confirmed as poor prognostic factors for recurrent disease. Stage II patients had remarkable post-recurrence survival compared to stage III patients. The recurrence interval was not found to be a prognostic factor for post-recurrence survival of colorectal cancer patients. Short (<1 year) or long (>4 years) intervals between resection and recurrence of the primary tumor were not associated with a poorer prognosis. In a country with limited resources, baseline tumor and patient characteristics provide clinically significant factors in defining individual prognosis of patients with recurrent CRC.

## Figures and Tables

**Figure 1 medicina-57-01108-f001:**
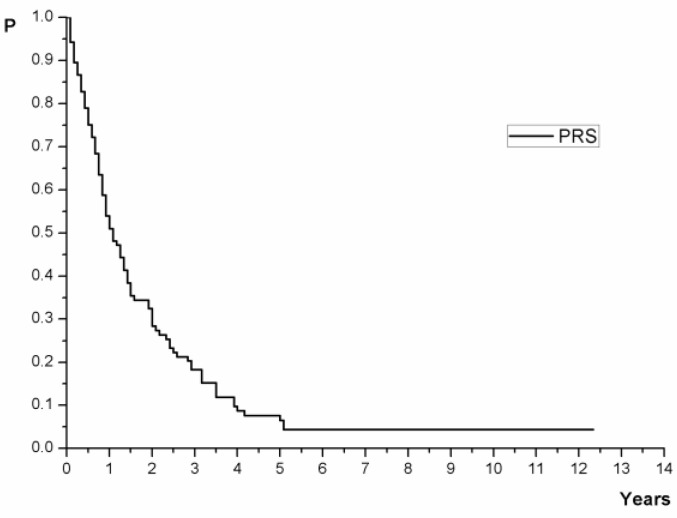
Post-recurrence survival for all patients.

**Figure 2 medicina-57-01108-f002:**
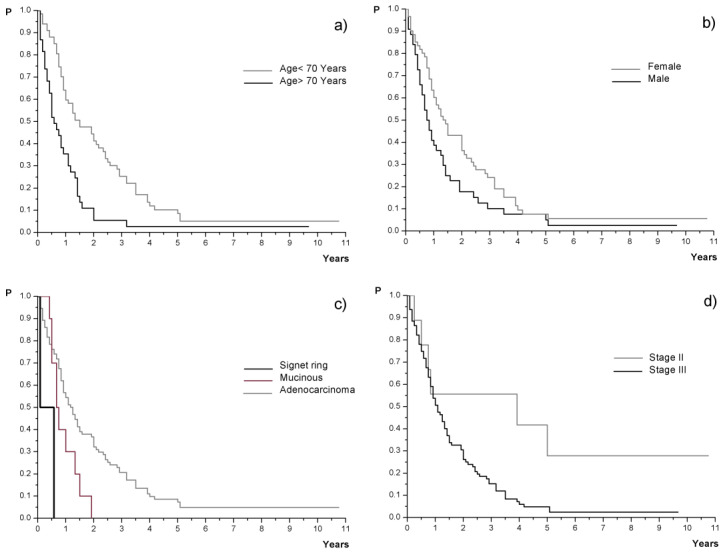
Kaplan–Meier curves for post-recurrence survival according to (**a**) age at diagnosis, *p* < 0.01; (**b**) gender, *p* < 0.05; (**c**) histology, *p* < 0.01; (**d**) clinical stage at diagnosis, *p* < 0.05.

**Figure 3 medicina-57-01108-f003:**
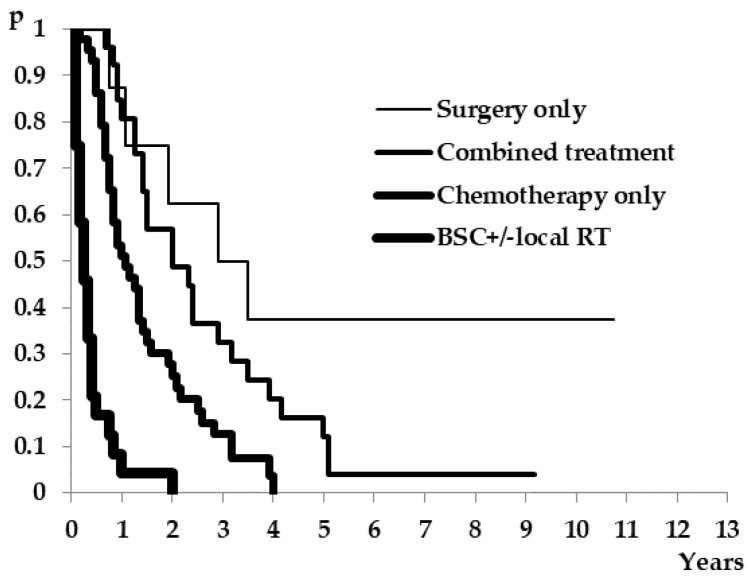
Kaplan–Meier curves for post-recurrence survival according to salvage treatments.

**Table 1 medicina-57-01108-t001:** Cumulative percentage of yearly post-recurrence survival.

Time	No in Risk	Cumulative Percentage with 95% CI
1 year	56 (53.33%)	51.04 (42.28–61.60)
2 year	32 (30.48%)	28.37 (20.85–38.60)
3 year	20 (19.05%)	18.24 (12.06–27.60)
4 year	9 (8.57%)	8.69 (4.54–16.60)
5 years	7 (6.67%)	6.51 (3.04–13.90)

**Table 2 medicina-57-01108-t002:** Patient and disease characteristics and influence on post-recurrence survival.

Characteristic	*N* (%)	Survival (Months)		Characteristic	*N* (%)	Survival (Months)	
		Median (95% CI)	Log-Rank Test			Median (95% CI)	Log-Rank Test
Age (years)				Differentiation			
Mean (SD)	65.6 (10)	/	/	Low grade	94 (89.5%)	13 (11–17)	ns
Median (Range)	67 (20–84)			High grade	11 (10.48%)	11 (≥6)	
Age (categories)				Lymphatic invasion			
<70 yrs	67 (63.8%)	18 (12–29)	<0.01	Yes	73 (69.5%)	7.5 (≥5)	ns
≥70 yrs	38 (36.2%)	7 (5–13)		No	14 (13.4%)	15 (11–18)	
Gender				No data	18 (17.1%)	/	
Male	44 (41.9%)	9.5 (7–16)	<0.05	Vascular invasion			
Female	61 (58.1%)	17 (12–25)		Yes	44 (41.9%)	11 (9–30)	ns
Localization				No	35 (33.3%)	14 (9–18)	
Right side	36 (34.3%)	10 (8–14)	ns	No data	26 (24.8%)	/	
Left side	69 (65.7%)	16 (12–24)		T stage			
Bowel obstruction				T2	4 (3.8%)	13 (≥2)	ns
Yes	33 (31.4%)	13 (10–29)	ns	T3	84 (80%)	13 (11–18)	
No	72 (68.6%)	13 (10–18)		T4a	3 (2.9%)	23 (≥5)	
Bowel perforation				T4b	14 (13.3%)	9 (6–50)	
Yes	12 (11.4%)	13.5 (≥10)	ns	N stage			
No	93 (88.6%)	13 (10–18)		N0	10 (9.5%)	28.5 (≥6)	ns
ECOG PS †				N1a	12 (11.4%)	20.5 (≥11)	
PS0	89 (84.8%)	12 (10–17)	ns	N1b	24 (22.9%)	16 (11–34)	
PS1	16 (15.2%)	16.5 (6–48)		N2a	24 (22.9%)	13 (10–24)	
CEA				N2b	33 (31.4%)	9 (7–19)	
≤5 mg/mL	86 (81.9%)	12 (10–18)	ns	No data	2 (1.9%)	/	
>5 ng/mL	16 (15.2%)	14.5 (5–42)		Stage			
No data	3 (2.9%)	/		II	9 (8.6%)	47 (≥9)	<0.05
Ca19-9				III	96 (91.4%)	13 (10–17)	
≤37 U/mL	91 (86.7%)	15 (11–18)	<0.01	First recurrence			
>37 U/mL	9 (8.6%)	8 (≥4)		Local	18 (17.1%)	20.5 (10–42)	ns
No data	5 (4.7%)	/		Systemic disease	11 (10.5%)	8 (≥5)	
Adjuvant CHT				Combined	76 (72.4%)	12.5 (10–17)	
5FU-LV	37 (35.2%)	23 (11–34)	ns	Early recurrence			
Capecitabine	68 (64.8%)	11 (9–16)		<1 year	39 (37.1%)	11 (10–19)	ns
Histology				≥1 year	66 (62.9%)	13 (10–23)	
Adenocarcinoma	93 (88.6%)	15 (11–19)	<0.01	Late recurrence			
Mucinous	10 (9.5%)	8.5 (≥6)		≥4 years	13 (12.4%)	10 (≥6)	ns
Signet ring	2 (1.9%)	4 (≥1)		<4 years	92 (87.6%)	13.5 (12–18)	

**Table 3 medicina-57-01108-t003:** Univariate and multivariate analyses of prognostic factors for post-recurrence survival.

Parameters	Univariate Cox Regression		Multivariate Cox Regression	
	HR (95% CI)	Wald Test	HR (95% CI)	Likelihood Ratio Test
Age ≥70 yrs vs. <70 yrs	2.19 (1.42–3.38)	*p* < 0.01	2.43 (1.55–3.81)	*p* < 0.01
Gender Male vs. Female	1.45 (0.96–2.19)	ns	-	
Histology Mucinous vs. Adenocarcinoma Signet ring vs. Adenocarcinoma	2.07 (1.02–4.20) 6.73 (1.58–28.58)	*p* < 0.05 *p* < 0.05	1.51 (0.73–3.10) 9.69 (2.23–42.0)	
Clinical stage Stage III vs. Stage II	2.64 (1.13–6.17)	*p* < 0.05	-	
Ca19–9 >37 U/ml vs. ≤37 U/mL	2.71 (1.33–5.52)	*p* < 0.05	3.51 (1.68–7.37)	

**Table 4 medicina-57-01108-t004:** Salvage treatments and their influence on survival.

Salvage Treatment	*N* (%)	Median Survival (Months) 95%CI	Log-Rank Test
BSC +/− local RT	24 (22.8%)	3 (2–5)	<0.01
Surgery only	8 (7.6%)	38.5 (23-NR)	
Chemotherapy only	41 (39%)	13 (10–19)	
Combined treatment	24 (22.9%)	24 (17–42)	

## Data Availability

Data available on request due to privacy restrictions.

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
