# Peer review of "Prognostic Factors for Post-Recurrence Survival in Stage II and III Colorectal Carcinoma Patients"

_medicina, 2021, doi:10.3390/medicina57101108_

Round 1

Reviewer 1 Report

The research article of Neda Nikolic et al., entitled “Post-recurrence survival in stage II and III colorectal carcinoma patients-evaluation of real-world prognostic factors”, evaluates the prognostic factors of post-recurrence survival in local and locally advanced colorectal cancer patients, and highlights the importance of the particular study, as the number of studies on prognostic factors of the post-recurrence survival is quite small. It is a well-structured and well-presented article with interesting results. However, I have a few minor concerns:

  • The title of the manuscript must be rephrased. I believe that the adjective “real-word” is not suitable for characterizing the prognostic factors, as it creates to the reader a misleading for the credibility of the existing prognostic factors.
  • The introduction appears to be quite modest. The authors are advised to add more information regarding CRC in the introduction section, in order to describe the biological background of the present disease sufficiently.
  • The references are not sufficient. The authors are advised to enrich the manuscript’s “References” section (relevant to the previous comment).
  • English correction is needed. The main text contains several grammatical and syntax errors that must be corrected.

Author Response

Comments and Suggestions for Authors

The research article of Neda Nikolic et al., entitled “Post-recurrence survival in stage II and III colorectal carcinoma patients-evaluation of real-world prognostic factors”, evaluates the prognostic factors of post-recurrence survival in local and locally advanced colorectal cancer patients, and highlights the importance of the particular study, as the number of studies on prognostic factors of the post-recurrence survival is quite small. It is a well-structured and well-presented article with interesting results. However, I have a few minor concerns:

1. The title of the manuscript must be rephrased. I believe that the adjective “real-word” is not suitable for characterizing the prognostic factors, as it creates to the reader a misleading for the credibility of the existing prognostic factors.

Answer: We thank the reviewer for the constructive comments. The Title has been rephrased accordingly to:

“Prognostic factors for post-recurrence survival in stage II and III colorectal carcinoma patients”

The corresponding sentence has been deleted from the discussion as well.

2. The introduction appears to be quite modest. The authors are advised to add more information regarding CRC in the introduction section, in order to describe the biological background of the present disease sufficiently.

Answer: The Introduction section has been reinforced with additional background data as suggested (page 2 line 2-13 and line 24-28 of the revised manuscript)

3. The references are not sufficient. The authors are advised to enrich the manuscript’s “References” section (relevant to the previous comment).

Answer: The References section has been updated accordingly to correspond to the updated Introduction section (page 10 references no. 5-10, 13, 14, page 11 references 19, 20, 30 of the revised manuscript)

4. English correction is needed. The main text contains several grammatical and syntax errors that must be corrected.

Answer: The manuscript has been thoroughly edited and changes have been introduced throughout the revised manuscript using the track changes option.

Reviewer 2 Report

The Authors report on a retrospective observational study investigating prognostic factors and survival in stage II and III colorectal cancer patients submitted to surgical excision and post-operative systemic therapy and experiencing disease relapse. This is a monocentric study, with long term follow up. Few comments:

  • Authors state: ‘recurrence was categorized as local, distant and combined. Please clearly define the pattern of recurrence, in paticularl what authors considered as local. How were regional (nodal) recurrence classified?
  • Results: line 132. Please use gender, instead of sex.
  • I think part of the eventual differences in survival data may have been influenced by salvage therapies. I would suggest to present the data about salvage therapies and explore eventual differences in survival depending on the type of salvage treatment.

Author Response

Comments and Suggestions for Authors

The Authors report on a retrospective observational study investigating prognostic factors and survival in stage II and III colorectal cancer patients submitted to surgical excision and post-operative systemic therapy and experiencing disease relapse. This is a monocentric study, with long term follow up. Few comments: 

1. Authors state: ‘recurrence was categorized as local, distant and combined. Please clearly define the pattern of recurrence, in paticular what authors considered as local. How were regional (nodal) recurrence classified?

Answer: We thank the reviewer for the constructive comments. The pattern of recurrence has been defined and data on the classification of regional (nodal) recurrence introduced as suggested (page 3, line 19-22 of the revised manuscript)

2. Results: line 132. Please use gender, instead of sex.

Answer: The term “sex” has been replaced with “gender” throughout the revised manuscript

3. I think part of the eventual differences in survival data may have been influenced by salvage therapies. I would suggest to present the data about salvage therapies and explore eventual differences in survival depending on the type of salvage treatment.

Answer: Data on salvage therapies has been introduced in the Methodology section 2.3. Salvage therapies (page 3, line 22 of the revised manuscript) and their effect on survival explored and presented in the Results section (page 7, line 1-11 and on table 4 and figure 3 of the revised manuscript) and the data was discussed in the discussion section (page 8, line 24-38)

Round 2

Reviewer 2 Report

I am here thanking the authors for implementing the changes that were suggested. I do not have any further comment.